# Exploring Source View Capability: Improve Generalizable 3D Reconstruction with Multi-view Context from Source Views

## Abstract

Recent generalizable 3D reconstruction methods have been facing challenges in constructing geometry-consistent 3D features. This is primarily due to the source views conveying redundant information to the sampled 3D points that they do not observe, resulting in the samples struggling to distinguish the correct observations of them. We attribute this issue to that canonical supervision methods focus solely on the rendered target view from a single viewpoint, overlooking source views that capture the scene from different perspectives. With this insight, we pioneer a supervision method for source views, which can be applied alongside existing target view supervision in each iteration. Specifically, we define the Learned Geometry of the Scene (LGS) as source-view depth distributions, which are derived from the weights of source views for each sampled 3D point. To regularize the LGS to better model the real-world geometry, we introduce a novel unsupervised learning objective, which mitigates the optimization bias in existing objectives and ensures the LGS is more concentrated near the real-world geometry surface. Regularizing the LGS effectively helps filter out irrelevant source views for each sampled 3D point, and thus noticeably improves the performance of backbones. Mathematical proof is provided to validate the proposed objective, and extensive experiments demonstrate that our supervision method significantly improves both NeRF- and 3DGS-based backbones with negligible computation overhead.

## 1 Introduction

Starting with Neural Radiance Fields (NeRF) (Mildenhall et al., 2021), vast amount of per sence optimization methods (Liu et al., 2020; Barron et al., 2022; Kerbl et al., 2023; Yu et al., 2024) spring out as the protagonist of 3D reconstruction in recent years. However, these methods overfit to each scene and can not generalize to new ones, limiting their possible applications. As an alternative, generalizable reconstruction methods synthesize novel views on the fly of few source views without per-scene optimization (Wang et al., 2021; Charatan et al., 2024). These methods typically aggregate visual features extracted from source views to produce 3D feature for each sampled 3D point, and render each target-view pixel by integrating the feature of 3D points projecting on it.

Though there has been adequate research for generalizable reconstruction methods (Yu et al., 2021; Wang et al., 2021; Charatan et al., 2024; Wewer et al., 2024), they continue to struggle with constructing precise 3D features for sampled 3D points because of including irrelevant source views during the sample-wise feature aggregation, which hinders target view rendering. As shown in Figure 1a, irrelevant source views may prevent the 3D feature from paying attention to correct views, or mislead it to denote an occupied space. This problem is derived from the absence of the geometric constraint, that a source-view image should be only relevant to the 3D points lying around the observed geometry surface. Meeting this constraint requires 3D reasoning to determine the visibility of a 3D point from various viewpoints. To tackle this problem, most methods adopt a self-adaptive manner to satisfy the geometric constraint implicitly, which either utilize costly 3D-CNN to introduce global 3D context (Johari et al., 2022; Gu et al., 2020; Chang et al., 2022; Liu et al., 2022) to refine the 3D features, or powerful modules such as transformers for source-view aggregation (Ren et al., 2023; Wang et al., 2022; Suhail et al., 2022; Chen et al., 2024). Recent methods take one step forward by explicitly

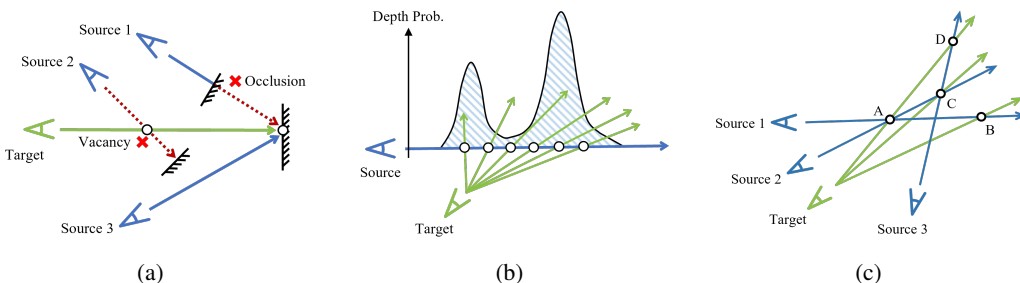

Figure 1: **(a) Example of irrelevant source views hindering sample-wise source-view aggregation.** For the sampled 3D point on the surface, source view 1 may prevent its feature from paying attention to source view 3. For the sampled 3D point in the vacancy, source view 2 provides information of a solid surface, which may mislead its feature to denote an occupied space. **(b) The discrete depth distribution of a source ray.** We register the samples collected on target rays to source rays. After registration, the value of a sample is the weight of the source view for it. **(c) Supervising the samples from different source views is 3D-context-aware** by engaging samples which are not on the same plane. Consider plane ABC and ACD as not overlapped, and thus point A, B, C, D compose a 3D tetrahedron. Simultaneously constraining source rays engages A, B, C, D mediately.

predicting the visibility of the sampled 3D points to source views to guide source-view aggregation (Huang et al., 2023; Liu et al., 2022).

Different from all works above, we explore a brand new approach to satisfy the geometric constraint. We propose to utilize the geometric constraint to regularize the Learned Geometry of the Scene (LGS), which we define as depth distributions of source-view pixels. Specifically, with the widely used weighted pooling mechanism for source-view aggregation (Wang et al., 2021; Liu et al., 2022; Charatan et al., 2024; Liu et al., 2024), we denote the visibility of the sampled 3D points to source views as weights in weighted pooling, and transform the them into a discrete depth distribution for each source-view pixel ray (shorten as *source rays* in the following discussion), as illustrated in Figure 1b. By regularizing the depth distributions to be pulse-like through an unsupervised learning objective, the geometric constraint can be satisfied without any geometric ground truth.

However, for the regularization of the discrete depth distribution, we find that the optimization bias of existing unsupervised learning objective (Barron et al., 2022) will lead to a gap between real-world geometry and the optimized LGS. This gap behaves as the pulse of the optimal distribution for the objective occurs tightly at certain sample, rather than the actual hitting surface of the ray (depicted in Figure 3b, and discussed in Section 3.3). To mitigate the optimization bias, we devise a novel unsupervised learning objective to regularize the discrete depth distribution, whose optimal solutions compose a field that contains the real-world distribution, so that the object surface can be modeled in a regressive manner continuously, as depicted in Figure 3c.

The simultaneous supervision with source views from crossed perspectives engages the sampled 3D points by multi-view 3D context, as illustrated in Figure 1c. Benefiting from directly supervising the source-view aggregation process, our proposed supervision is algebraically interpretable, and can be carried alongside existing target view supervision with one target view rendering in a plug-and-play manner. To summarize our contributions in a few words:

- To the best of our knowledge, we are the first to propose source-view supervision for generalizable 3D reconstruction, which can be carried along target view supervision without additional rendering. Our supervision method filters out irrelevant source views for the sampled 3D points by regularizing the Learned Geometry of the Scene.

- We devise a novel unsupervised learning objective to regularize the discrete depth distributions of source-view pixels to be pulse-like, which mitigates the optimization bias by ensuring its optimal solutions to form a field that contains the real-world distribution.

- We provide mathematical proof to verify the devised learning objective, and extensive experiments demonstrate our proposed supervision method correctly encourages the confidence of the LGS, thereby improving the performance of both NeRF- and 3DGS-based backbones.

## 2 RELATED WORKS

**3D Reconstruction and Its Generalization.**   3D reconstruction, building the scene based on several known views and corresponding camera parameters, has been long investigated as an important topic in computer vision. Recently proposed NeRF (Mildenhall et al., 2021) and a mass of follow-up studies (Barron et al., 2021; Chen et al., 2022; Hu et al., 2023; Barron et al., 2022; 2023; Kerbl et al., 2023; Yu et al., 2024) have drawn the community's attention to per-scene optimized reconstruction methods. Though there has already been methods that support fast per-scene training and rendering (Wizadwongsa et al., 2021; Müller et al., 2022; Barron et al., 2023; Kerbl et al., 2023), they typically still require minutes to train and densely captured known views, blocking their possible applications. Meanwhile, inspired by NeRF, researchers start to combine radiance field with epipolar geometry (Wang et al., 2021; Yu et al., 2021; Charatan et al., 2024) to form a new paradigm, which is so called generalizable reconstruction. Alternative to per-scene optimized methods, generalizable methods learn the common sense of vision and geometry to adapt to different scenes. Recent generalizable reconstruction methods tend to apply transformer (Vaswani et al., 2017) in their networks to fuse the feature of sampled 3D points and source views (Suhail et al., 2022; Ding et al., 2022; Ren et al., 2023), some of which even completely discard multi-layer perceptron (MLP) based volume rendering (Wang et al., 2022). On the other hand, fully image-to-image methods propose to predict novel views in a geometry-free manner (Sajjadi et al., 2022; Kulhánek et al., 2022). This paper follows the canonical 3D reconstruction framework (Wang et al., 2021), which builds understanding of the scene with epipolar geometry.

**Multi-View 3D Context.**   Awareness of multi-view 3D context is obliviously important for 3D reconstruction. While early neural representation based works straightly embody the multi-view context with epipolar geometry (Yu et al., 2021; Chen et al., 2021; Wang et al., 2021), recent works inspired by surface reconstruction methods tend to refine the volume features with 3D-CNN (Ding et al., 2022; Chang et al., 2022; Peng et al., 2022; Huang et al., 2023; Liu et al., 2024; Chen et al., 2024). On the other hand, image-to-image methods propose to learn the 3D context in the embedding space with a completely implicit manner (Sajjadi et al., 2022; Kulhánek et al., 2022). Different from these works, we believe that source views could be utilized more than simply as input, thus we propose to supervise the LGS derived from source views with geometric constraint. With simultaneous constraint from different viewpoints, the 3D context is explicitly embedded into the proposed supervision without any costly 3D module like 3D-CNN, as depicted in Figure 1c.

**Source-View Geometry in Generalizable Reconstruction.**   The most related works to this paper are NeuRay (Liu et al., 2022) and its follow-up works (Huang et al., 2023), who try to explore the geometric information of source views to constrain the generalizable reconstruction model. For example, NeuRay assumes that the visibility of the sampled 3D points on the same source-view ray follows the probability density function of a logistics distribution, and predicts the shape factors of the logistics distribution to infer the weights of source views to samples explicitly. Such an approach requires a specific network design, which performs well but cannot directly benefit other existing generalizable reconstruction methods. However, our work is significantly distinct from NeuRay and its follow-up works. What we propose is **not** a specific network design, but a plug-and-play supervision method that improves the performance of both NeRF- and 3DGS-based generalizable reconstruction backbones. Moreover, we propose a novel objective to mitigate the optimization bias of the existing objective (Barron et al., 2022), which is the key of our method but has not been discussed by existing works to the best of our knowledge.

## 3 METHOD

**Overview.** We present **S**ource-view **G**eometric **C**onstraint (SGC), a novel source-view based supervision method to regularize the source-view *encoder* of generalizable reconstruction backbones to better model the real-world geometry, as the result improving the performance of backbones *decoding* the scene with either NeRF or 3DGS. We first introduce the pipeline of epipolar geometry based generalizable reconstruction methods in Section 3.1. Next, we present how SGC is applied to these backbones and improves their performance. Specifically, we explain how the LGS is efficiently derived from a given backbone in Section 3.2, and then we introduce how to regularize the LGS and mitigate the optimization bias issue with our devised unsupervised objective in Section 3.3.

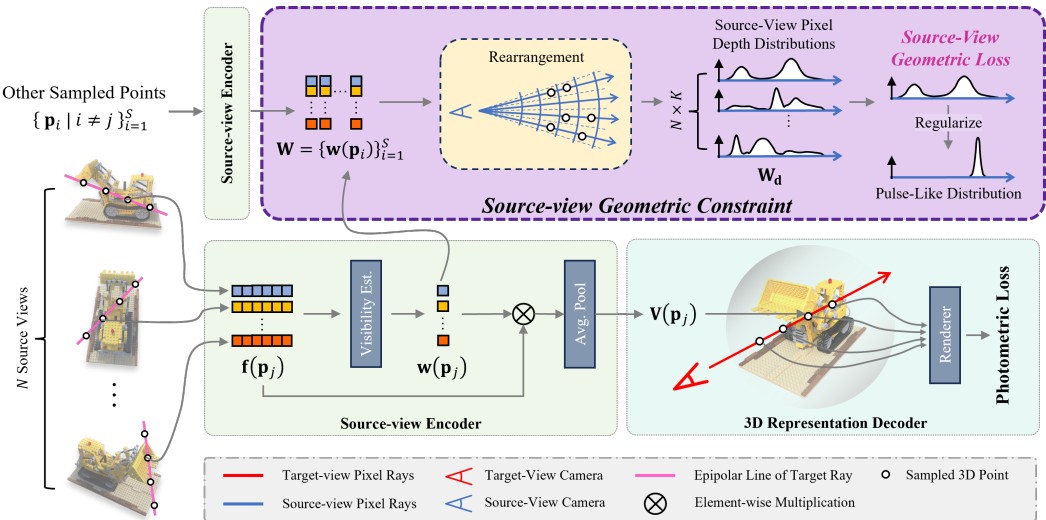

Figure 2: **Pipeline of our source-view supervision method.** Our method directly supervises the source-view encoder of generalizable reconstruction methods with geometric constraint. Canonically, to encode source views and represent the scene, 3D points sampled in the space are projected to all source views with epipolar geometry for extracting local visual features. The visual features are then aggregated to produce 3D feature of the samples through weighted pooling. To apply our supervision method, we regard the weights as the visibility of the samples to source views, derive the depth distribution of source-view pixels from the weights with an efficient rearrangement algorithm and regularize the depth distributions to be pulse-like with our devised source-view geometric loss.

### 3.1 EPIPOLAR GEOMETRY BASED GENERALIZABLE 3D RECONSTRUCTION

Generalizable reconstruction methods render an unknown target view $\mathbf{I}_0 \in \mathbb{R}^{H \times W}$ based on $N$ known source views $\{\mathbf{I}_i \in \mathbb{R}^{H \times W}\}_{i=1}^N$ and corresponding camera parameters $\{\mathbf{P}_i \in \mathbb{R}^{3 \times 4}\}_{i=0}^N$. Typically, the pipeline of generalizable reconstruction methods could be broken down as *encoder* and *decoder*, where the encoder fuse the visual features from source views to build a representation of the scene, and the decoder resolves the scene representation to produce novel observations.

Take ray casting based rendering pipelines (Wang et al., 2021; Liu et al., 2024) for example. As shown in Figure 2, rays $\{\mathbf{r}_j = \mathbf{o}_0 + t\mathbf{d}_j | \mathbf{o}_0, \mathbf{d}_j \in \mathbb{R}^3, t \in \mathbb{R}\}_{j=1}^{H \times W}$ are cast from target viewpoint $o_0$ (shorten as *target rays* in the following discussion) to imitate the observation of each target-view pixel. For each target ray, $n$ samples $\{\mathbf{p}_j^k = \mathbf{o}_0 + t_k\mathbf{d}_j \in \mathbf{r}_j\}_{k=1}^n$ are collected and projected to source views by epipolar geometry to extract visual features. Formally, given $\mathbf{f}_{\mathrm{src}}(\mathbf{p}) = [\mathbf{f}_1(\mathbf{p}), ..., \mathbf{f}_N(\mathbf{p})]^T$ denoting the corresponded visual features of sample $\mathbf{p} \in \mathbb{R}^3$ in $N$ source views, a source-view aggregation mechanism is typically adopted,

$$\mathcal{M}_{\mathrm{encode}} : \left(\mathbf{f}_{\mathrm{src}}(\mathbf{p}), \{\mathbf{P}_i\}_{i=0}^N\right) \mapsto \mathbf{V}(\mathbf{p}), \tag{1}$$

to obtain the 3D feature $\mathbf{V}(\mathbf{p})$ of sample $\mathbf{p}$ by aggregating source-view features, where $\mathbf{V}(\mathbf{p})$ is the encoded 3D representation of the scene, capable for embedding both appearance and geometry. Subsequently, sampling based rendering is performed by fusing samples on each target ray to decode the pixel color $c(\mathbf{r}_j)$, which could be denoted as,

$$\mathcal{M}_{\mathrm{decode}} : \left\{\mathbf{V}(\mathbf{p}_j^k) | \mathbf{p}_j^k \in \mathbf{r}_j\right\}_{k=1}^n \mapsto c(\mathbf{r}_j). \tag{2}$$

### 3.2 THE LEARNED GEOMETRY OF THE SCENE

Canonically, generalizable reconstruction methods are supervised with target-view losses, which constrain samples on the same target ray to produce the right color and other properties such as depth. Since a ray is one-dimensional, supervision from a single viewpoint only embodies 1D context for

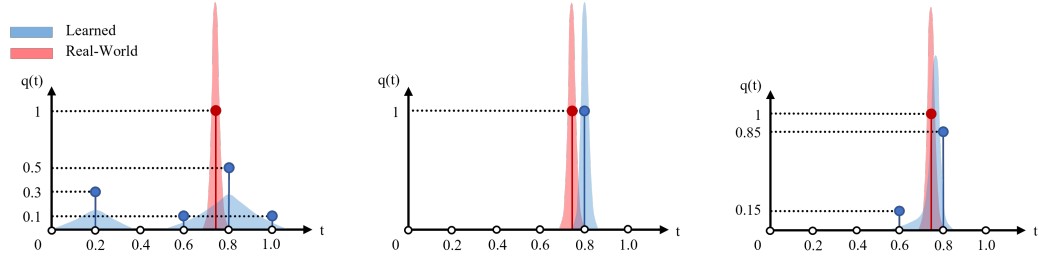

(a) The learned depth distribution of the source rays is not regularized.

(b) The optimal learned depth distribution regularized with optimization bias.

(c) The optimal learned depth distribution regularized without optimization bias.

Figure 3: Blue/red denotes the learned/real-world geometry, and stem/line denotes corresponded discrete/continuous depth distribution. **(a) If the learned depth distribution is not regularized**, its density can be scattered, indicating the network doesn't properly understand the real-world geometry. **(b) The learned depth distribution is regularized with** $\mathcal{L}_{\text{dist}}$ (Barron et al., 2022), which introduces optimization bias. The gap between the optimal and real-world depth distribution indicates a strong geometric misunderstanding. **(c) We mitigate the optimization bias** by allowing the optimal solution of $\mathcal{L}_{\text{sgc}}$ to regress the real-world surface with adjacent samples. Mitigating optimization bias indicates the optimal solution of the regularization objective can model the real-world better.

the samples on the same target ray. The lack of 3D context in supervision leads to the weakness in reasoning the visibility of the sampled 3D points to source views. To tackle this problem, we propose to supervise the LGS with SGC. Concluding SGC in a sentence, we rule the discrete depth distribution of each source-view pixel to be pulse-like.

There are multiple means to obtain the source-view depth distributions. The most straight forward one is to carry additional rendering on source views with difference samples for each view. However, this can lead to $N$ times of computation overhead compared to a single target-view rendering. Such an approach is infeasible for generalizable reconstruction models, which typically take days to train.

To this end, we propose a novel approach to the distributions by using the intermediate results during target-view rendering. As shown in Figure 2, to derive the LGS of a given backbone for a specific scene, we first denote the visibility of a sampled 3D point $\mathbf{p}$ to $N$ source views as the weights $\mathbf{w}(\mathbf{p}) = [w_1(\mathbf{p}), ..., w_N(\mathbf{p})]^T \in \mathbb{R}^{N \times 1}$ of weighted pooling (Wang et al., 2021) for source-view aggregation, where $w_i(\text{p})$ denotes how much source view $i$ contributes to $\mathbf{V}(\mathbf{p})$. With $S$ samples collected for representing the scene in total, a visibility matrix $\mathbf{W}_{\text{v}} = [\mathbf{w}(\mathbf{p}_1), ..., \mathbf{w}(\mathbf{p}_S)] \in \mathbb{R}^{N \times S}$ could be built as the explicit representation of the LGS. To regularize $\mathbf{W}_{\text{v}}$, we have to rearrange $\mathbf{W}_{\text{v}}$ into depth distributions of source-view pixels, which we denote as $\mathbf{W}_{\text{d}} \in \mathbb{R}^{(N \times H \times W) \times L}$, where $L$ indicates the uniform length of all $N \times H \times W$ depth distributions.

However, the rearrangement is non-trivial since samples collected on target rays leads to irregular sampling frequency of source rays, as a result of which $L$ is inconsistent for source rays, resulting in unfriendly computation for GPU. To tackle this problem, we propose to rearrange $\mathbf{W}_{\text{v}}$ by the depth-bins (the dashed cells in Figure 2) on each source ray, where the value of a bin indicates its visibility to the source view, and each source ray shares the same number of bins. To save the page for the coming discussion in Section 3.3, we suggest reader to further details in Appendix A.3.1. Thanks to the efficient rearrangement algorithm, we can now obtain the depth distributions of each source ray alongside the rendering of target view with negligible computation overhead. In the next section, we are going to introduce how to regularize the discrete depth distributions $\mathbf{W}_{\text{d}}$.

### 3.3 DISCRETE DEPTH DISTRIBUTION REGULARIZATION

#### 3.3.1 DISCRETE DISTRIBUTION TO BE PULSE-LIKE

With $L$ samples $\{\mathbf{p}_k\}_{k=1}^L$ of a source ray $\mathbf{r}_s = \mathbf{o}_i + t\mathbf{d}_s$ cast from source viewpoint $\mathbf{o}_i$, where $t \in [a, b], i = 1, ..., N, s = 1, ..., N \times H \times W$, the discrete depth distribution of $\mathbf{r}_s$ is defined as,

$$\mathcal{Q}(\mathbf{r}_s) : q(\mathbf{p}_1), q(\mathbf{p}_2), \ldots, q(\mathbf{p}_L), \tag{3}$$

where $q(\mathbf{p}_i)$ denotes the probability that the surface hit by $\mathbf{r}_s$ lies around $\mathbf{p}_i$. With $\mathbf{W}_\mathrm{d}$ introduced in Section 3.2, we can straightly obtain $\mathcal{Q}(\mathbf{r}_s)$ from any line of it. By default we suppose $\mathcal{Q}(\mathbf{r}_s)$ regularized as a probability density function (PDF) that $q(\mathbf{p}_i)$ are non-negative and sum up to 1.

Ideally, with $L \to \infty$, $\mathcal{Q}(\mathbf{r}_s)$ should be a pulse distribution whose peak occurs at the surface hit by $\mathbf{r}_s$. It could be easily regularized with the objective proposed by Barron et al. (2022),

$$\mathcal{L}_\mathrm{dist} = \frac{1}{3} \sum_{i=1}^{L} q_i^2 (x_i - x_{i-1}) + \sum_{i=1}^{L} \sum_{j=1}^{L} q_i q_j |\frac{x_i + x_{i-1}}{2} - \frac{x_j + x_{j-1}}{2}|, \tag{4}$$

where $q_i$ denotes $q(\mathbf{p}_i)$ for briefness, $a = x_0 < \cdots < x_L = b$ and $[x_{i-1}, x_i)$ denotes the interval containing $\mathbf{p}_i$. In the following discussion, we suppose the samples as linearly collected, by which we have $x_i = a + (b - a)\frac{i}{L}$. In this case, $x_i$ represents just the relative instead of the actual position of the sample. We further exam this assumption in Appendix A.2.

### 3.3.2 SOURCE-VIEW GEOMETRIC CONSTRAINT OBJECTIVE

**Optimization Bias from Optimal Condition.** Despite $\mathcal{Q}(\mathbf{r}_s)$ could be easily regularized in the ideal case, $L$ is far smaller than $\infty$ in practice due to computational limitation, in which case Equation 4 would introduce optimization bias. More concretely, Barron et al. (2022) achieved Equation 4 from its continuous form with the assumption that $q_i$ is consistent within each sample interval $[x_{i-1}, x_i)$. This is not satisfied in practice where $\mathcal{Q}(\mathbf{r}_s)$ is discrete and sparse and thus being the root of the optimization bias. Denoting the interval length as $l = x_i - x_{i-1} = (b - a)/L$, Equation 4 could be reformed equivalently into,

$$\mathcal{L}_\mathrm{dist} = \frac{l}{3} \left[ \left( \sum_{i=1}^{L} q_i \right)^2 + \sum_{i=1}^{L} \sum_{j=1}^{L} q_i q_j |i - j| + 2 \sum_{i=1}^{L} \sum_{j=1}^{L} [|i - j| \geq 2] q_i q_j (|i - j| - 1) \right], \tag{5}$$

where $\sum_{i=1}^{n} q_i$ is **constant** to be 1 for that $\mathcal{Q}(\mathbf{r}_s)$ is regularized as a PDF, and $[|i - j| \geq 2] = 1$ if $|i - j| \geq 2$ and 0 otherwise. The optimal value of Equation 5 is $l/3$, which is achieve if and only if the last two terms in the square brackets are 0, or in other words, only one $q_i$ is non-zero. As Figure 3b illustrates, such an optimal solution introduces geometric gap between the LGS and real-world geometry, because real-world surface rarely lies tightly to certain sample.

**Mitigate Optimization Bias with Relaxed Optimal Condition.** With the optimal solution of Equation 5 unraveled, a solution to mitigate the optimization bias is straight forward. We modify the optimal condition of Equation 5 to: there are at most two non-zero samples and they must be adjacent to each other. With the new optimal condition, the real-world surface could be better approximated with the continuous regression of adjacent samples on source rays, as depicted in Figure 3c. The optimal value of the two non-zero samples $q_i$ is left to the existing objectives like photometric loss to decide. The last term of Equation 5 meets the new optimal condition well, while it's the second term that introduces the optimization bias who reaches 0 only when there is at most one non-zero $q_i$. If we directly remove the second term of Equation 5, it can be rewritten as,

$$\mathcal{L}_\mathrm{sgc} = \frac{l}{3} \left[ \left( \sum_{i=1}^{L} q_i \right)^2 + \alpha \sum_{i=1}^{L} \sum_{j=1}^{L} [|i - j| \geq 2] q_i q_j (|i - j| - 1) \right], \tag{6}$$

whose optimal condition is the one we devise, and $\alpha$ denotes a weight factor equalling 2 in Equation 5. Scaling up the whole objective by 3, Equation 6 can be further reformed as,

$$\mathcal{L}_\mathrm{sgc} = l \left[ \sum_{i=1}^{L} q_i^2 + \sum_{i=1}^{L} \sum_{j=1}^{L} q_i q_j |i - j| + (\alpha - 1) \sum_{i=1}^{L} \sum_{j=1}^{L} [|i - j| \geq 2] q_i q_j (|i - j| - 1) \right]. \tag{7}$$

Taking $\alpha = 1$ to remove the last term of Equation 7 for simplified computation, we adopt $\mathcal{L}_\mathrm{sgc}^{\alpha=1}$ to regularize $\mathcal{Q}(\mathbf{r}_s)$, which we refer to as the SGC loss in the following discussion.

For thorough understanding of Section 3.3, we strongly suggest the readers to Appendix A.1 and A.2. The former provides another approach to devise $\mathcal{L}_\mathrm{sgc}^{\alpha=1}$, and the latter exams the linear sampling assumption mentioned in Section 3.3.1. Furthermore, in Section 4.3 we present experimental evidence about why the optimal condition is important for the LGS regularization objective.

Table 1: **Qualitative results of the NeRF-based backbone with SGC supervision.** With GNT (Wang et al., 2022) being the backbone, our SGC supervised GNT is denoted as GNT+SGC($\mathcal{L}_{sgc}$), while GNT+SGC($\mathcal{L}_{dist}$) denotes replacing $\mathcal{L}_{sgc}$ with $\mathcal{L}_{dist}$. The † indicates the result directly referenced from published papers (Wang et al., 2021; Liu et al., 2022; Cong et al., 2023).

| | NeRF Synthetic | | | | Local Light Field Fusion (LLFF) | | | |
|---|---|---|---|---|---|---|---|---|
| | PSNR↑ | SSIM↑ | LPIPS↓ | Avg↓ | PSNR↑ | SSIM↑ | LPIPS↓ | Avg↓ |
| IBRNet† | 26.73 | 0.908 | 0.101 | 0.040 | 25.17 | 0.813 | 0.200 | 0.064 |
| NeuRay† | **28.29** | 0.927 | 0.080 | 0.032 | 25.35 | 0.818 | 0.198 | 0.062 |
| GNT | 27.25 | 0.935 | 0.059 | 0.030 | 25.88 | 0.863 | 0.123 | 0.049 |
| GNT-MOVE† | 27.47 | **0.940** | 0.056 | 0.029 | **26.02** | **0.869** | **0.108** | **0.043** |
| GNT+SGC($\mathcal{L}_{dist}$) | 27.31 | 0.935 | 0.057 | 0.030 | 25.85 | 0.861 | 0.125 | 0.049 |
| GNT+SGC($\mathcal{L}_{sgc}$), ours | 27.87 | 0.939 | **0.054** | **0.027** | 25.96 | 0.867 | 0.119 | 0.047 |

Table 2: **Qualitative results of the 3DGS-based backbone with SGC supervision.** With pixelSplat (Charatan et al., 2024) being the backbone, metrics of both small and large baseline of source views are listed, with +SGC($\mathcal{L}_{dist}$) denoting supervising the backbone with the distortion loss proposed by Barron et al. (2022), and +SGC($\mathcal{L}_{sgc}$) denoting supervising the backbone with our proposed SGC loss. Experiments are conducted on the DTU dataset. See discussion in Section 4.2

| | Small Baseline | | | | Large Baseline | | | |
|---|---|---|---|---|---|---|---|---|
| | PSNR↑ | SSIM↑ | LPIPS↓ | Avg↓ | PSNR↑ | SSIM↑ | LPIPS↓ | Avg↓ |
| pixelSplat | 26.88 | 0.777 | **0.197** | 0.058 | 21.68 | 0.645 | 0.334 | 0.111 |
| +SGC($\mathcal{L}_{dist}$) | 27.20 | 0.778 | 0.199 | 0.057 | 21.87 | 0.647 | 0.333 | 0.109 |
| +SGC($\mathcal{L}_{sgc}$), ours | **27.27** | **0.783** | 0.198 | **0.056** | **22.02** | **0.650** | **0.333** | **0.107** |

## 4 EXPERIMENT

**Overview.** We conduct experiments on both NeRF- and 3DGS-based backbones to perform how the proposed SGC supervision improves their performance. Section 4.1 and 4.2 presents experiments on NeRF- and 3DGS-based backbones, respectively. And we discussion why the convergence condition of the regularization objective is vital (and other findings) from the curve of the LGS regularization objective throughout the training span in Section 4.3. Ablations study is provided in Appendix A.4.

### 4.1 NERF-BASED BACKBONE

**Implementation Details.** For the NeRF-based backbone, we choose GNT (Wang et al., 2022), a fully transformer-based network that utilizes subtraction efficient attention (Zhao et al., 2021) to encode source views for each sample, and decodes the target view with transformer fusing the features on the target rays. We believe such a full transformer network is representative as a NeRF-based generalizable reconstruction method. We attach $\mathcal{L}_{sgc}^{\alpha=1}$ with a weight of $10^{-3}$ to the existing loss and train the network from scratch. Further experimental details could be found in Appendix A.3.3.

**Datasets and Metrics.** Following Wang et al. (2022), we perform training on 7 indoor and outdoor, realistic and synthetic datasets: Google Scanned Object (Downs et al., 2022), RealEstate10K (Zhou et al., 2018), the Spaces dataset (Flynn et al., 2019), and handheld phone captures (Mildenhall et al., 2019). For evaluation, NeRF Synthetic (Mildenhall et al., 2021) and LLFF (Mildenhall et al., 2019) datasets are adopted. We report average metrics across all scenes, including Peak Signal-to-Noise Ratio (PSNR), Structural Similarity Index Measure (SSIM) (Wang et al., 2004), and the Learned Perceptual Image Patch Similarities (LPIPS) (Zhang et al., 2018). The geometric mean of $10^{-PSNR/10}$, $\sqrt{1 - SSIM}$ and LPIPS is also reported as a summary of the three metrics.

**Result.** As shown in Table 1, GNT+SGC($\mathcal{L}_{sgc}$) significantly outperforms the vanilla GNT. What's more, on the NeRF Synthetic dataset, GNT+SGC($\mathcal{L}_{sgc}$) even visibly surpasses GNT-MOVE (Cong et al., 2023), the upgraded version of GNT which equips the mixture-of-export inspired by large language models and an additional spatial consistency loss at the same time. Our proposed SGC

supervision is concise, elegant while being more effective than GNT-MOVE, indicating the superiority of our method. To intuitively see how the SGC supervision improves GNT, we show evaluation renderings in Figure 4. It's clear to see that GNT tends to produce defective rendering with hollows on objects or blurs around edges, which are significantly improved with our SGC supervision. Meanwhile, replacing $\mathcal{L}_{sgc}$ with $\mathcal{L}_{dist}$ yields no difference of performance from the vanilla GNT, which strongly advocates the importance of $\mathcal{L}_{sgc}$. Depth renderings in Figure 5 intuitively express how the defects in color rendering are formed.

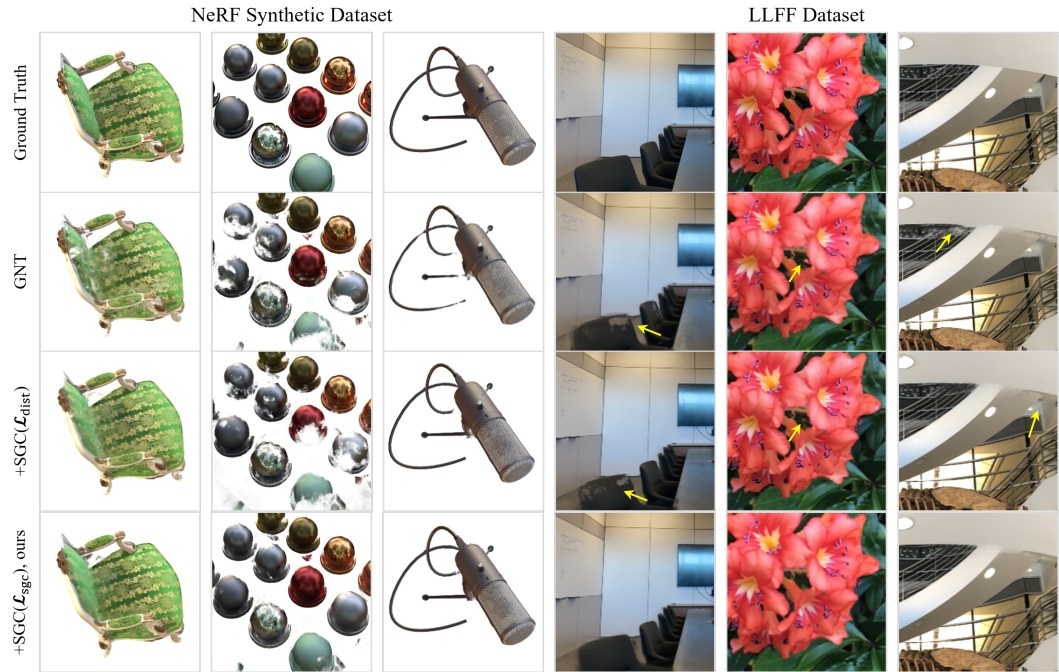

Figure 4: **Rendering results of NeRF-based experiments**, where GNT is our NeRF-based backbone, +SGC($\mathcal{L}_{dist}$) denotes applying SGC to the backbone with the distortion loss (Barron et al., 2022), and +SGC($\mathcal{L}_{sgc}$) denotes applying SGC to the backbone with our proposed SGC loss. Yellow arrows in the last 3 columns are devoted to point out the alleviated local defects.

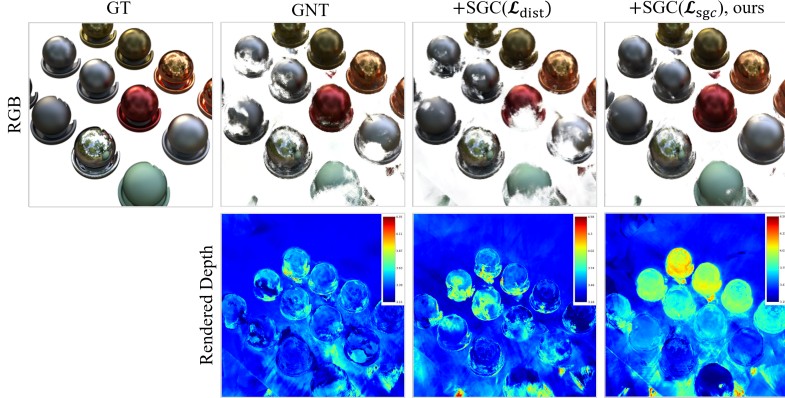

Figure 5: **Depth rendering results of NeRF-based experiments**, visualized by the the weight-averaged position of samples on each target ray. The depth and color rendering echos well, where GNT and GNT+SGC($\mathcal{L}_{dist}$) produce hollow in the corresponding region of both domains, while our method produces well rendered color and smoothed depth. The results demonstrate how our SGC supervision improves GNT by equipping it with a precise sense of geometry.

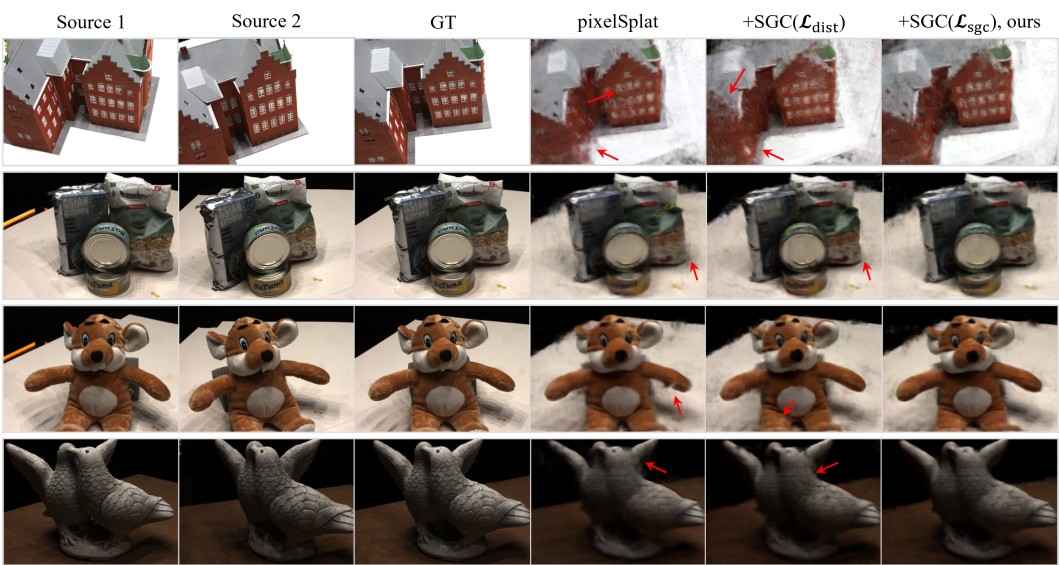

| Source 1 | Source 2 | GT | pixelSplat | +SGC($\mathcal{L}_{\text{dist}}$) | +SGC($\mathcal{L}_{\text{sgc}}$), ours |

Figure 6: **Rendering results of 3DGS-based experiments.** Case names are kept consistent with Table 2. The SGC supervised pixelSplat (either implemented with $\mathcal{L}_{\text{dist}}$ or $\mathcal{L}_{\text{sgc}}$) succeeds to help render sharper and more reasonable results compared to the vanilla pixelSplat.

### 4.2 3DGS-BASED BACKBONE

**Implementation Details.** We have pixelSplat (Charatan et al., 2024), the most representative 3DGS-based generalizable reconstruction method as the 3DGS-based backbone. Specifically, for encoding source views, pixelSplat adopts attention mechanism for each pixel with the pixels on its epipolar line in the other view, which is a different paradigm from GNT. We apply our SGC loss on the intermediate attention vectors that embeds depth information, and attach the SGC loss by a weight of $10^{-3}$ to the existing objectives. More experimental details can be found in Appendix A.3.4.

**Datasets and Metrics.** Since the RealEstate10k dataset (Zhou et al., 2018), which is the original dataset for training pixelSplat in Charatan et al. (2024), has been used for training the NeRF-based backbone in this paper, we perform experiments for the 3DGS-based backbone on the challenging DTU dataset (Jensen et al., 2014) in order to expand the field of evaluation data for our method. The partition of training and evaluation sets follows canonical multi-view stereo works (Yao et al., 2018). We use the same evaluation metrics as Section 4.1.

**Results.** We show qualitative results in Table 2. The SGC enhanced backbones significantly improves over the vanilla one in both small and large baseline settings, where we define the small baseline as a source-view index distance of 2 (as DTU camera moves like a snake), and the large baseline as the source-view index distance of 6.

**Discussion.** It's worth noting that replacing $\mathcal{L}_{\text{sgc}}$ with $\mathcal{L}_{\text{dist}}$ has a more slight impact on pixelSplat than GNT. We attribute this to the different sampling strategies adopted by the two methods for encoding the scene, that GNT samples on 3D target rays, while pixelSplat samples on the epipolar lines in the 2D images. As the result the former has extremely irregular sampling frequency of source rays, and a big portion of the source rays cannot be applied with SGC supervision for having too little samples. As contrary, the latter has uniformed sampling frequency for source rays and all the source-view pixels can be applied with SGC supervision. Irregular sampling is more likely to produce large sampling intervals, thus aggravates the optimization bias problem.

### 4.3 CONVERGENCE OF THE LGS REGULARIZATION OBJECTIVE

We present the curves of the LGS regularization objective under different experiments throughout the training span in Figure 7a (for GNT, $\mathcal{L}_{\text{sgc}}$ is depicted). The convergence of the curve is a good criterion to measure if the LGS suffers from aliasing because of including irrelevant source views.

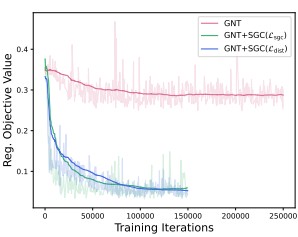
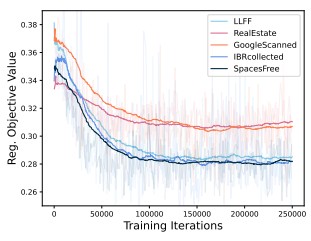
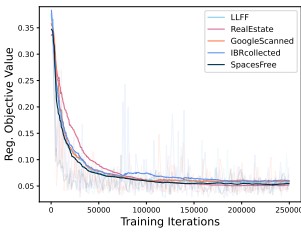

(a) Objective curves of 3 GNT based experiments.     (b) Dataset-wise curves for GNT     (c) Dataset-wise curves for GNT+SGC($\mathcal{L}_{\text{sgc}}$)

Figure 7: **The curves of the regularization objective throughout the training span.** (a) The objective curves of GNT, GNT+SGC($\mathcal{L}_{\text{dist}}$) and GNT+SGC($\mathcal{L}_{\text{sgc}}$) respectively. All the curves converge in the early stage of training. (b) The dataset-wise objective curves for GNT. There is significant gap between the converged value if difference datasets. (c) The dataset-wise objective curves for GNT+SGC. The objectives converges to a consistent value on all training datasets.

In Figure 7a, for the vanilla GNT, the curve converges at a high position, while as comparison, the curves converge to a much lower position when the regularization objective is supervised. On the other hand, the curves for GNT+SGC($\mathcal{L}_{\text{dist}}$) and GNT+SGC($\mathcal{L}_{\text{sgc}}$) in Figure 7a demonstrate that the convergence condition is achieved indeed during training, so different convergence condition of the regularization objective will surely affect the network behavior. This experimental result advocates why we should care about the convergence condition of the regularization objective.

Furthermore, we depict the dataset-wise curves of GNT and GNT+SGC($\mathcal{L}_{\text{sgc}}$) in Figure 7b and 7c, respectively. It is worth noting that curves of GNT behaves different convergence position, indicating there is gap between how GNT understands the scene over different datasets. As contrary, curves of GNT+SGC($\mathcal{L}_{\text{sgc}}$) converges to the same position, which indicates our SGC supervision encourage GNT to understand the scenes in a more generalized and uniformed approach.

## 5 DISCUSSION

The paper has meticulously demonstrated how the proposed SGC supervision method improves the performance generalizable 3D reconstruction methods. However, there are still few drawbacks of this work. First, as the experiments have demonstrated, the optimal condition of the regularization objective counts for the SGC supervision to have a positive impact to the backbone. Our proposed SGC loss doesn't really have its optimal solution to be the real-world geometry, but allows its optimal solution field to contain the real-world geometry. More precise solutions could be explored for the regularization objective. Second, as the regularization objective is unsupervised, it should not converge too early when the backbone don't have an overall 3D reasoning ability. Otherwise the unsupervised learning objective may mislead the backbone, resulting in degraded performance. We control the speed of convergence by a small weight for the objective, even so the objective converges very fast as shown in Figure 7a. Further research could be made on a more elegant solution to this issue. And we leave how SGC impacts surface reconstruction to the future.

## 6 CONCLUSION

In this paper, we innovatively propose Source-view Geometric Constraint, a supervision method which improves generalizable 3D reconstruction methods by regularizing the discrete depth distributions on source views. We further propose a novel unsupervised training objective to regularize the discrete depth distributions to be pulse-like, which crucially mitigates the gap between the optimized LGS and real-world geometry. The proposed supervision can be elegantly attached to the existing training objectives for a given backbone with negligible computation overhead, visibly improves the performance of both NeRF- and 3DGS-based backbones. Mathematical analysis and experimental results convincingly profile the mechanism of our work. We suggest the readers to the Appendix for further details and ablation study. We hope that this work exhibits a new direction to the community for improving generalizable reconstruction quality by supervision on source-view information.

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

# A APPENDIX

## A.1 ANOTHER APPROACH TO $\mathcal{L}_{\text{sgc}}^{\alpha=1}$

In this section, we demonstrate the uniqueness of $\mathcal{L}_{\text{sgc}}^{\alpha=1}$ if we try to achieve it by adjusting weights of terms in Equation 5. To make the discussion more clear, we first transform Equation 5 into an equivalent but simplified form,

$$\mathcal{L}_{\text{dist}} = l \left[ \frac{1}{3} \sum_{i=1}^{L} q_i^2 + \sum_{i=1}^{L} \sum_{j=1}^{L} q_i q_j |i - j| \right] \tag{8}$$

Let's consider the moment where only two $q_i$ are non-zero while being adjacent. As already discussed, the optimization bias comes from Equation 8 converging to a distribution where their is only one non-zero $q_i$, thus one of the two non-zero samples would be optimized to be zero. If we try to stop this trend to mitigate the optimization bias by adjusting ratio of the terms in the square brackets of Equation 8 to form a new objective $\mathcal{L}_b$, that

$$\mathcal{L}_b^{L=2} = l \left[ \lambda \sum_{i=1}^{L} q_i^2 + \sum_{i=1}^{L} \sum_{j=1}^{L} q_i q_j |i - j| \right] = l \left[ (\lambda - 1)(q_1^2 + q_2^2) + (q_1 + q_2)^2 \right], \tag{9}$$

then $\mathcal{L}_b^{L=2}$ should be consistent with any $q_1, q_2 > 0$ that sum up to 1 (regularized as PDF), where $\lambda > 0$ is a ratio factor controlling the ratio of the two terms. It's obvious that only $\lambda = 1$ can keep $\mathcal{L}_b^{L=2}$ being a constant. As the result, $\mathcal{L}_b^{\lambda=1} = \mathcal{L}_{\text{sgc}}^{\alpha=1}$ when $L = 2$.

To conclude, $\lambda = 1$ is a necessary condition for $\mathcal{L}_b$ to mitigate the optimization bias, leading to $\mathcal{L}_b^{\lambda=1} = \mathcal{L}_{\text{sgc}}^{\alpha=1}$ for any positive integer $L \geq 1$. As the convergence of $\mathcal{L}_{\text{sgc}}^{\alpha=1}$ has already been discussed in Section 3.3.2, we do not make further discussion on the convergence of $\mathcal{L}_b^{\lambda=1}$.

## A.2 LINEAR SAMPLING ASSUMPTION

As we mentioned in Section 3.3.1, we have a strong assumption that $x_i$ is linearly sampled regardless of the actual position of the sampled 3D points. Similar to Appendix A.1, with a discussion of the case where $L = 2$, it is trivial to prove that the optimization bias always exists for non-linearly sampled $x_i$. To mitigate the optimization bias, its the most straight forward approach to treat $x_i$ as linearly sampled.

## A.3 IMPLEMENTATION DETAILS

### A.3.1 REARRANGEMENT

Typically, for epipolar geometry based generalizable reconstruction methods, 3D points are sampled on target rays, resulting in irregular sampling frequency on source rays. Meanwhile, to correctly denote $\mathcal{Q}(\mathbf{r}_s)$, samples on the same source ray should be sorted by their depth to the corresponding source view. These two issues make it challenging to efficiently rearrange $\mathbf{W}_v$ into $\mathbf{W}_d$, from a GPU-unfriendly structure to a GPU-friendly structure, where the latter is the explicit representation of the LGS to be supervised.

To tackle the problem, we propose a devoted rearrangement algorithm. The algorithm utilize camera extrinsics and intrinsics to obtain view-specific depths and the projected pixels for each sampled 3D point, and then rearrange the samples with depth-bin based bucket sort.

**Depth-bin Partition.** In one target-view rendering pass, for each source-view pixel indexed $s = 1, ..., N \times H \times W$, we compute the depth range of samples projected to it, which is denoted as $[d_{\min}^s, d_{\max}^s)$. We linearly divide the range as $N_{\text{bin}}$ depth-bins for each source-view pixel, with the depth range of the $i$-th bin denoted as $[d_{i-1}^s, d_i^s)$, where $d_i^s = d_{\min}^s + i/N_{\text{bin}} \times (d_{\max}^s - d_{\min}^s)$.

**Register samples to the depth-bins.** With view-specific depth, the sampled 3D points could be registered to the depth-bins efficiently, with the value of the depth-bin being the weight of the specific source view for the sampled 3D point. Once the registering is complete, samples projected to the same source-view pixel are naturally ordered by depth. There are two advantages of the bucket sort, one is that the irregular sampling frequency of source rays are aligned with each other by the same bucket resolution $N_{\mathrm{bin}}$, the other is that it doesn't need any comparison between the sorted elements but only index registering, which is more efficient than sorting on GPU.

**Post Process.** After the bucket sort, for a group of bins of the same source-view pixel, we further bubble the non-empty bins to the front, pad the value of empty bins at the end with 0, and regularize depth bins of the same pixel to sum up to 1. Picking $K$ source rays with the highest sampling frequency for each of the $N$ source views and concatenate them together, we have $\mathbf{W}_{\mathrm{d}} \in \mathbb{R}^{(K \times N) \times N_{\mathrm{bin}}}$ as the explicit representation of (Section 3.2).

**Discussion.** The value of $N_{\mathrm{bin}}$ decides the probability of conflict where two samples are registered to the same depth-bin. This probability goes negligible as $N_{\mathrm{bin}}$ is large enough. More concretely, $N_{\mathrm{bin}}$ is equivalent to the $L$ in Section 3.2. On the other hand, the reason why we have $K$ is that, for a NeRF-based generalizable reconstruction method, only a batch of target-view pixels can be sampled in a single iteration because of out-of-memory issue. As the result, most of the source-view pixels only have little samples projected on them (usually less than 10 by our experiment). Such a sparse sampling can not depict a distribution well, which could even degrade performance of the backbone if used for supervision. So we only pick the $K$ most sampled pixels on each source view to perform depth distribution regularization.

### A.3.2 NORMALIZATION FOR $\mathcal{Q}(\mathbf{r}_s)$

In practice, we normalize $\mathcal{Q}(\mathbf{r}_s)$ by $q_i = q_i / (\epsilon + \sum q_j)$, where $\epsilon$ is a small positive constant. Since probably the real-world surface hit by $\mathbf{r}_s$ might be far from all sampled 3D-points on it, such a normalization enables $\mathcal{L}_{\mathrm{sgc}}^{\alpha=1}(\mathcal{Q}(\mathbf{r}_s))$ to be optimized as 0 in this case, where $q_i = 0$ for any $i$.

### A.3.3 ADDITIONAL DETAILS FOR NERF-BASED EXPERIMENTS

**SGC for GNT.** GNT (Liu et al., 2022) is a 8-layer transformer based network, which performs the aggregation-rendering loop for 8 times. For source-view aggregation, GNT utilize the subtraction efficient transformer (Zhao et al., 2021), which produces an attention vector $\mathbf{w}(\mathbf{p}) \in \mathbb{R}^{N \times 1}$ for each sampled 3D point $\mathbf{p}$, indicating its visibility to $N$ source views. To apply our SGC supervision to GNT, we derive $\mathbf{W}_{\mathrm{d}}$ by the averaging and rearrangement (Section 3.2) of $\mathbf{w}(\mathbf{p})$ in the first 4 layers, leaving the last 4 layers unchanged to remedy the possible misguide from the unsupervised learning objective. For the experiments in Section 4.1, we take $N_{\mathrm{bin}} = 1024$ and $K = 256$ for the computation of $\mathbf{W}_{\mathbf{d}}$.

**Training Details.** All NeRF-based experiments are performed on 8 Nvidia V100 GPUs. For the vanilla GNT, we follow Liu et al. (2022) to train for 250k iterations. For the GNT supervised by SGC, we only train for 150k iterations because we found its already enough for the network to converge in this case. Other training details, such as the learning rate and hyper-parameters are kept strictly consistent with Liu et al. (2022).

### A.3.4 ADDITIONAL DETAILS FOR 3DGS-BASED EXPERIMENTS

**SGC for pixelSplat.** Similar to GNT, pixelSplat (Charatan et al., 2024) utilizes a 2-layer transformer based network. We apply the SGC loss to regularize the attention vector from the epipolar attention mechanism. More concretely, only the attention vector in the first layer is regularized, and the second layer is left unchanged. And the rearrangement is not necessary for pixelSplat because of it's not adopting a ray casting based sampling strategy.

**Training Details**. All 3DGS-based experiments are performed on 2 RTX 3090 GPUs, trained for 200k iterations for convergence, with source-view baseline gradually increase in the first 10k steps. All hyper-parameters are directly referenced from Charatan et al. (2024) without any further adjustment for this paper. For difference light levels in the DTU dataset, light levels from 1 to 6 are adopted for training, while evaluation is conducted on the light level of 3.

Table 3: **Ablation studies on GNT.** Both settings achieve consistent conclusion that the proposed supervision improves rendering quality, and our $\mathcal{L}_{\text{sgc}}$ do alleviates the optimization bias.

| | NeRF Synthetic Dataset | | | LLFF Dataset | | |
|---|---|---|---|---|---|---|
| | PSNR↑ | SSIM↑ | LPIPS↓ | PSNR↑ | SSIM↑ | LPIPS↓ |
| (1) w/o SGC | 27.25 | 0.935 | 0.059 | 25.88 | 0.863 | 0.123 |
| (2) +SGC($\mathcal{L}_{\text{dist}}$) | 27.31 | 0.935 | 0.057 | 25.85 | 0.861 | 0.125 |
| (3) +SGC($\mathcal{L}_{\text{sgc}}$), $\lambda = 3$ | 26.90 | 0.934 | 0.057 | 25.74 | 0.863 | 0.121 |
| (4) +SGC($\mathcal{L}_{\text{sgc}}$), w/o top $K$ | 27.55 | 0.935 | 0.058 | 25.71 | 0.859 | 0.128 |
| (5) +SGC($\mathcal{L}_{\text{sgc}}$), ours | **27.87** | **0.939** | **0.054** | **25.96** | **0.867** | **0.119** |

## A.4 ABLATION STUDY

We conduct the following ablation studies on GNT to profile the key to the SGC supervision, the results are listed in Table 3.

**(1) Without SGC supervision.** We present the vanilla implementation of GNT (Liu et al., 2022) as the baseline of ablation. Also, the vanilla implementation of pixelSplat (Charatan et al., 2024) is presented in Table 2.

**(2) With SGC supervised by the distortion loss.** We present the backbone with SGC supervision but replacing our SGC loss with the distortion loss (Barron et al., 2022), and also this ablation for pixelSplat is presented in Table 2. For GNT, it comes out that there is no much difference and even significant degrade from the vanilla backbone. For pixelSplat, the ablation result has been discussed in the discussion paragraph of Section 4.2. The results strongly advocate out analysis in Section 3.3, that the optimization bias will mislead the backbone from benefiting from the SGC supervision.

**(3) Simply Manipulating $\lambda$.** As discussed in Appendix A.1, for $\mathcal{L}_b$, $\lambda$ must be 1 to mitigate the optimization bias. In this ablation case, we set $\lambda$ to be 3, which suffers from the optimization bias and record the result. As shown in Table 3, their is significant degrade from the vanilla GNT, evidences the analysis about the optimization bias with a different perspective from ablation case 2.

**(4) Randomly Supervising Source Rays.** As discussed in Appendix A.3.1, we find it important to have the most sampled source rays to be supervised by our method. In this ablation case, we pick the $K$ source rays randomly from each source view, rather than the $K$ most sampled ones, for our SGC supervision. Results come out that the impact is inconsistent on different evaluation datasets, and the performance is significantly worse than our top-$K$ implementation in ablation case 5.

**(5) Full Implementation of Our Method.** In ablation case 5, we present the full implementation of our method supervising GNT. The same result is also presented for pixelSplat in Table 2. The full implementation significantly outperforms all ablation cases. The ablation studies demonstrate the importance of sampling frequency of source rays for SGC supervision, the existence of the optimization bias, and the effectivity of our proposed SGC supervision and the devised objective.

## A.5 COMPUTATION OVERHEAD FOR REARRANGEMENT

In Table 4, we present the iteration-wise additional time for $\mathcal{L}_{\text{sgc}}^{\alpha=1}$ computation, mainly consists of the rearrangement time cost. The first line represents the same setting used in Section 4.1. The computation time is insensitive to $N_{\text{bin}}$, while linearly relates to $K$ because of unavoidable CPU operation in our implementation. The additional time cost for computing $\mathcal{L}_{\text{sgc}}^{\alpha=1}$ is less than 4% of the time for a single training step, having negligible burden on training.

Table 4: Computation time of $\mathcal{L}_{\text{sgc}}^{\alpha=1}$ with respect to $N_{\text{bin}}$ and $K$.

| $N_{\text{bin}}$ | $K$ | Time (ms) |
|---|---|---|
| **1024** | **256** | **24** |
| 1024 | 512 | 41 |
| 1024 | 1024 | 76 |
| 4096 | 256 | 27 |

