# OpenReview forum: "Exploring Source View Capability: Improve Generalizable 3D Reconstruction with Multi-view Context from Source Views"
_ICLR.cc/2025/Conference — ICLR 2025 Conference Withdrawn Submission_

### Official Review · Reviewer_veq7 · 2024-10-27

**Soundness:** 3
**Presentation:** 3
**Contribution:** 3
**Rating:** 5
**Confidence:** 4

**Summary:**

The paper proposes a novel regularization objective for generalizable 3D reconstruction from multiple source views. Prior work typically involves projecting sampled points onto multiple source views and then learning a set of weights for each source view to aggregate extracted features effectively. Building on this, the core idea of this paper is to learn these weights by regularizing their depth distribution -- computed directly from the sample points of target views -- to ensure it is unimodal and to save computation. This is achieved by relaxing the regularization constraint in MipNeRF360, allowing at least two adjacent samples with non-zero depth probability instead of restricting to a single sample. This approach demonstrates improvements in NeRF-based and 3DGS-based methods for novel view synthesis.

**Strengths:**

- The paper is well-written and clearly presented.
- The proposed idea of relaxing the regularization in MipNeRF360 is sound and simple to implement, making it adaptable to other rendering-based methods.
- Figure 3 effectively illustrates the motivation and impact of the proposed regularization.
- The mathematical derivation of the proposed objective is intuitive and easy to follow.

**Weaknesses:**

- The transition from Equation (6) to (7) appears incorrect. For example, the second term of Equation (5) is removed in Equation (6) but reappears in Equation (7). Additionally, the square of the sum in the first term becomes the sum of squares.
- When $\alpha$ in Equation (7) is set to 1.0 to remove the last term, it is unclear how this the result loss function $\mathcal{L}_{\rm sgc}=(\sum q_i)^2$ maintains the constraint of having at most two adjacent non-zero samples, although it holds for Equation (6).
- Improvements appear limited, as shown in Table 1, with gaps of only 0.62, 0.004, and 0.004 in terms of PSNR/SSIM/LPIPS on the NeRF synthetic dataset. Similar trends are observed on the LLFF dataset and on the comparisons to pixelSplat in Table 2.
- Qualitatively, as shown in Figure 4, the proposed constraint provides minimal improvement; only one scene (second column) out of six shows clear improvement, while the rest exhibit marginal gains. A similar pattern appears in Figure 6 for 3DGS baselines.
- Figure 5 seems to be a cherry-picked example, as it shows the most substantial improvement from Figure 4. Additional examples would better demonstrate the method's benefits.
- The paper presents itself as a 3D reconstruction method but only includes novel view synthesis results, lacking explicit 3D reconstruction results.

**Questions:**

- How are the weights for the regularization constraint set?
- How do the number of depth bins (and their sizes) and the number of source rays impact the results?
- Given that the source ray depth distribution is derived from target view sample points, many empty bins will likely exist along the source rays. How sensitive is the regularization objective to the number of empty bins?
- Would it be more convincing to include plots similar to Figure 3 but with the learned ray distribution?

---

### Official Review · Reviewer_bzo3 · 2024-11-01

**Soundness:** 4
**Presentation:** 3
**Contribution:** 3
**Rating:** 5
**Confidence:** 5

**Summary:**

The paper introduces Source-view Geometric Constraint (SGC), a new supervision method for generalizable 3D reconstruction that leverages multi-view context to improve 3D feature consistency. Key contributions include regularizing source-view depth distributions through an unsupervised pulse-like objective to reduce optimization bias. Extensive experiments show that SGC significantly enhances NeRF- and 3DGS-based methods with minimal computational overhead, demonstrating better geometry representation and scene understanding.

**Strengths:**

* Depth regularization on the source view is both intriguing and innovative in the field of generalizable 3D reconstruction, with well-founded motivation.
* The design, description, and reasoning of the "discrete depth distribution regularization" are clear and logical.
* The GNT-based and pixelSplat-based experiments partially provide foundational validation for the effectiveness and soundness of the proposed method (for details, please refer to the [Weaknesses]).

**Weaknesses:**

* **Limitations of the method**. In the experiments, regularization is applied based on the source view's attention map along the target ray/epipolar line. It appears that the proposed depth regularization may be directly compatible only with transformer-based generalizable NeRFs and generalizable 3DGS that use an epipolar transformer.
* **Insufficient comparative experiments**. As the authors mention, this work introduces a novel supervision method, SGC, to enhance the performance of generalizable 3D reconstruction. However, in the NeRF-based backbone experiments, SGC supervision is only added to one method, GNT [1]. I strongly recommend conducting experiments on more generalizable NeRF methods to validate the effectiveness of SGC, such as GNT follow-up methods like EVE-NeRF [2] and GNT-MOVE [3], as well as the transformer-based approach GPNR [4].
* **Regarding the concern of "converge too early."** In L522, "it should not converge too early when the backbone doesn't have an overall 3D reasoning ability," the solution proposed is to "control the speed of convergence by a small weight." Would it be possible to introduce SGC midway through training? Adding relevant experiments and analysis could be beneficial.

[1] Wang P, Chen X, Chen T, et al. Is Attention All That NeRF Needs?[J]. arXiv preprint arXiv:2207.13298, 2022.

[2] Min Z, Luo Y, Yang W, et al. Entangled View-Epipolar Information Aggregation for Generalizable Neural Radiance Fields[C]//Proceedings of the IEEE/CVF Conference on Computer Vision and Pattern Recognition. 2024: 4906-4916.

[3] Cong W, Liang H, Wang P, et al. Enhancing nerf akin to enhancing llms: Generalizable nerf transformer with mixture-of-view-experts[C]//Proceedings of the IEEE/CVF International Conference on Computer Vision. 2023: 3193-3204.

[4] Suhail M, Esteves C, Sigal L, et al. Generalizable patch-based neural rendering[C]//European Conference on Computer Vision. Cham: Springer Nature Switzerland, 2022: 156-174.

**Questions:**

Kindly refer to the [Weaknesses].

---

### Official Review · Reviewer_a2ic · 2024-11-03

**Soundness:** 2
**Presentation:** 3
**Contribution:** 2
**Rating:** 5
**Confidence:** 4

**Summary:**

This paper introduces a new method called Source-View Geometric Constraint (SGC), which enhances the generalization and geometric consistency of 3D reconstruction models by supervising the depth distribution of source views. The SGC method can be integrated with target view supervision without additional rendering and incorporates an unsupervised learning objective to reduce optimization bias, ensuring the model better aligns with real 3D geometry.

**Strengths:**

1.	It proposes source-view geometric constraints for generalizable 3D reconstruction, combined with target view supervision, to enhance geometric consistency.
2.	It designs an unsupervised objective for regularizing depth distributions, reducing optimization bias and aligning the model more closely with real-world geometry.

**Weaknesses:**

1.	The improvement over existing work is marginal. The method deals with very similar issue as NeuRay, i.e. how to aggregate the samples in a ray according to their visibility or importance to the novel views. However, when compared with NeuRay, the improvement is very marginal or worse on some metrics. Also, why the comparison on DTU dataset in Table 1 is missing?
2.	The ablation of adding the proposed loss on the GNT backbone in Table 1 also demonstrates the very marginal improvement, making it suspectable as evidence of the proposed method's effectiveness. Though in Table 2, in large baseline, the method demonstrates large improvements over pixelSplat, it does not show whether on GNT, similar improvements can be achieved. Especially, compared with NeuBay, can the method demonstrate similar improvements?
3.	To consolidate the work, IBRNet+ SGC and NeuRay+SGC are suggested.

**Questions:**

See weakness

---

### Official Review · Reviewer_Rj9v · 2024-11-04

**Soundness:** 3
**Presentation:** 3
**Contribution:** 2
**Rating:** 3
**Confidence:** 3

**Summary:**

The paper proposes a novel depth regularization objective as geometry supervision for generalizable 3D reconstruction tasks. The depth distributions of the source-view pixels are estimated by predicting the visibility of scene sample points to the source views. These distributions are then regularized to be pulse-like, thereby reducing erroneous correspondences between source and target views.  The paper further optimizes the previously proposed depth regularization objective by relaxing the optimal condition to allow two adjacent non-zero samples on a source ray instead of just one. The novel regularization term can improve reconstruction quality and training efficiency of both NeRF- and 3DGS-based backbones.

**Strengths:**

1.	The analysis of the problem is clear, and the proposed method is reasonable and targeted.
2.	The proposed SGC loss is easy to implement and does not require any additional input or output.
3.	SGC loss can eliminate some undesirable artifacts, enhance geometric consistency and improve training efficiency.

**Weaknesses:**

1.	The technical novelty is not sufficient, as the key technical contribution, which is a change in the optimal condition of depth regularization proposed by Mip-Nerf 360, lacks significant innovations.
2.	The improvement of the method is limited. Although this method produces better reconstruction results in certain areas, it does not achieve results comparable to the baseline in other areas. As shown in Fig. 6, the roof of the house is more blurred compared to PixelSplat, suggesting unstable improvement.
3.	The experiments in the paper are insufficient. The paper should include more combinations with different baselines to demonstrate the stability of quality improvements better. The paper also conducts extensive analysis of the incorrect correspondence between viewpoints and proposes to supervise on-scene geometry. The paper should present results under more input-output scenarios and additional geometric visualization results to effectively demonstrate the method's effectiveness.

**Questions:**

See weakness 2.

---

### Note · Authors · 2024-11-18

**Comment:**

Thanks to all the reviewers' time and patience. Our work needs to be further refined to be published. After discussion among the coauthors, we decide to withdraw this paper. Sincerely thanks to the reviewers.

**Withdrawal Confirmation:**

I have read and agree with the venue's withdrawal policy on behalf of myself and my co-authors.